# Effects of Fence Enclosure on Vegetation Community Characteristics and Productivity of a Degraded Temperate Meadow Steppe in Northern China

**Lijun Xu [1], Yingying Nie [1], Baorui Chen [1], Xiaoping Xin [1],\*, Guixia Yang [1], Dawei Xu [1] and Liming Ye [2,3],\***

[1]  Chinese Academy of Agricultural Sciences, Institute of Agricultural Resources and Regional Planning; Hulunbuir Grassland Ecosystem Research Station, Beijing 100081, China; xulijun@caas.cn (L.X.); nieyingying0908@126.com (Y.N.); chenbaorui@caas.cn (B.C.); yangguixia@caas.cn (G.Y.); xudawei@caas.cn (D.X.)
[2]  Department of Geology, Ghent University, 9000 Gent, Belgium
[3]  Chinese Academy of Agricultural Sciences, Institute of Agricultural Resources and Regional Planning, Beijing 100081, China
\*  Correspondence: xinxiaoping@caas.cn (X.X.); liming.ye@ugent.be (L.Y.)

**Abstract:** Species composition and biomass are two important indicators in assessing the effects of restoration measures of degraded grasslands. In this paper, we present a field study on the temporal changes in plant community characteristics, species diversity and biomass production in a degraded temperate meadow steppe in response to an enclosure measure in Hulunbuir in Northern China. Our results showed that the plant community responded positively to the fence enclosure in terms of vegetation coverage, height, above- and belowground biomass. A year-to-year increase in aboveground biomass was observed, and this increase plateaued at the ninth year of the enclosure. Our results also showed that the existing dominant and foundation species gained predominance against other species. The sum of the biomass of these two species was more than doubled after the ninth year of the enclosure. However, belowground biomass only briefly increased until the fifth year of the enclosure and then decreased until the end of the experimental period. Plant diversity, evenness, and richness indices showed similar trends to that of belowground biomass. Overall, we found that the degraded temperate meadow steppe responded significantly positively to the enclosure treatment, but an optimal condition was only reached after approximately 5–7 years of continuous protection, providing a solid use case for grassland conservation and management at regional scales.

**Keywords:** temperate meadow steppe; grassland degradation; biomass; vegetation; community characteristics; enclosure; restoration; management; policy options

## 1. Introduction

Grassland provides multiple ecosystem functions in both natural and managed systems. It acts as an ecological barrier for, e.g., farmland systems against environmental infringement. It is also by itself a production base of quality livestock products that provide irreplaceable constituents for food and nutritional security at the global scale [1–4]. However, grassland degradation has extensively been observed in recent years due to over-exploitation [4–8]. The structure, function, and dynamics of grassland ecosystems have received increasing attention in the past few decades in China [8–10]. The grassland acreage in China accounts for $3.5 \times 10^6$ km$^2$, one-fourth of which is distributed in the Inner Mongolia region [4,11,12]. Located in northeastern Inner Mongolia, Hulunbuir possesses the most

concentrated and representative temperate meadow steppe in the whole of China, characterized by its native vegetation, rich biodiversity, and diversified landscapes [13,14]. However, the degradation of Hulunbuir grasslands has become more and more evident, and both technological and policy-level countermeasures are urgently needed [7,15–18]. The degradation rate in terms of percent area jumped from less than 10% during the 1980s and 1990s to more than 50% during 2000–2010. Grassland degradation has now become not only a hurdle for regional production but also a threat to the eco-environmental balance in wider areas across the country [14,19].

In a broader context, the grasslands of Inner Mongolia form an important part of the grasslands of Eurasia, with $7.9 \times 10^5$ km$^2$ of natural grassland. Common local practice of grassland management is characterized by a strict functional and spatial delineation between grazing and hay-harvesting [8]. Of the grasslands of Inner Mongolia, $6.4 \times 10^5$ km$^2$ or 81% is used for grazing, while $7.1 \times 10^4$ km$^2$ or 9% is mown for hay, mostly in the Hulunbuir area [12]. As the most important livestock industry region in China, Inner Mongolia owns a multi-year average stocking rate of $1 \times 10^8$ standard sheep units (SSU, derived from livestock numbers using a conversion rate of 1 for sheep; 0.8 for goats; and 5 for cattle, horses, and camels [20]). The mutton output of Inner Mongolia accounted for 22% of the national total output in 2018, while 18% of milk consumption in China was produced here [20]. However, high grazing pressure has led to severe degradation and desertification, with decreased forage production and environmental deterioration. Although systematic survey data are not available, remote sensing-assisted estimation suggests that the regional forage production decreased from $1.9 \times 10^8$ t in 2005 to $1.3 \times 10^8$ t in 2009 [4]. In contrast, actual stocking rate increased from $1.1 \times 10^8$ SSU in 2005 to $1.4 \times 10^8$ SSU in 2009, resulting in an increase in overgrazing rate from 23% to 112% during the same period [4]. Conversely, continued grassland degradation is expected to harm economic development as well as cause ecosystem instability at the regional to national scales [21]. By the end of the 20th century, 90% of the grassland condition of Inner Mongolia had been degraded to various extents [22], compared to an average of 22% at the national level [7]. This sharp contrast has stimulated growing attention to the protection and restoration of grasslands in recent decades.

Many scientists have engaged in combating grassland degradation. Progress has been made in many areas including rational grazing [7,23,24], enclosure [11,25,26], mowing [12,27,28], rapid planting [7,29,30], etc. Among others, fence enclosure is regarded as a simple yet effective measure [26,31] to increase vegetation coverage [30,32], improve species richness [33,34], raise productivity [35,36], and conserve soil water and nutrients [32,37].

Technically, the duration of the enclosure engagement is an important factor affecting the structure and diversity of plant species in grassland communities [30], especially in the moderately degraded pasture or heavily degraded grassland. An earlier study [38], conducted in a Hulunbuir desertified grassland, showed that longer enclosure duration tended to increase plant diversity and stability. However, some other studies suggested contrary effects of enclosure duration on plant community. For instance, Yan et al. [39] found that longer duration of enclosure reduced plant diversity in the neighboring Horqin desertified grassland. Although the effects of enclosure duration are being disputed, not only based on results from different types of grasslands, but also based on results from the same type of grassland [28], the enclosure itself has been recognized as an economical and effective measure to achieve self-healing and to rebuild ecosystem balance in degraded grasslands [8,26].

On the one hand, beneficial effects can usually be expected from an enclosure treatment. The enclosure creates a physical protection to the grassland from external disturbances, allowing the plants enough time to reproduce, which in turn results in an increased species diversity [40] and a higher chance to restore the plant communities [41]. On the other hand, the enclosure treatment may also give rise to adverse effects. As a matter of fact, enclosure is a human disturbance which eliminates grazing and trampling by large herbivores and blocks the exchange of energy and substances between the enclosed grassland and the external ecosystem [42]. Therefore, finding a balance between beneficial and adverse effects of an enclosure on the community structure and species diversity of grassland

ecosystems has become a hotspot of debate. From a practical perspective, finding such a balance also has great significance for the sustainable use of grasslands.

Therefore, we aim to quantitatively determine the optimal duration of the enclosure engagement for the restoration of a typical heavily degraded temperate meadow steppe of the region using field experiments in this research. More specifically, the objectives of this paper are: (1) to determine the effects of a fence enclosure on grassland productivity; (2) to evaluate plant community's responses to the enclosure treatment; (3) to determine the optimal duration of the enclosure treatment; and (4) to formulate technical options and policy recommendations for the restoration and sustainable use of the regional grassland resources. We fulfil these objectives by measuring how long it takes the degraded meadow steppe under the enclosure treatment to reach maximum biomass harvest on a yearly basis, and by analyzing the compositional changes in plant species of the meadow steppe vegetation in relation to the enclosure duration.

## 2. Materials and Methods

### 2.1. Site Description

The study was conducted at the Hulunbuir Grassland Ecosystem Observation and Research Station (49°19′35″N, 119°56′52″E) in northeastern Inner Mongolia, China (Figure 1). Temperate continental monsoon climate prevails in the region. Average monthly temperature varies from −48.5 °C in January to 36.2 °C in July, with a mean annual temperature of 1.5 °C. Active accumulated temperature of >10 °C is measured at 1700–2300 degree-days, which corresponds to a frost-free period of approximately 110 days. The mean annual precipitation ranges from 350 mm to 400 mm during 2000–2010, 80% of which falls in June to September [43]. The fenced area under the enclosure treatment is in a typical meadow steppe of the region, whose degradation class was evaluated as heavily degraded [22]. The major foundation species in the plant community is *Stipa baicalensis*, while the dominant species is *Leymus chinensis*, with a range of associated species, including *Cleistogenes squarrosa*, *Melissilus ruthenica*, *Carex duriuscula*, *Ixeris chinensis*, and some other weeds. Kastanozems is the dominant soil type in the region [44,45].

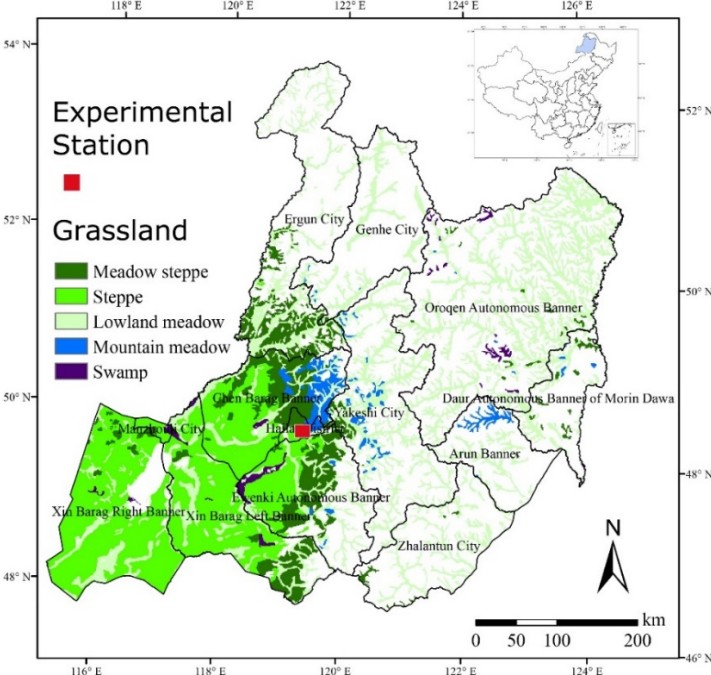

**Figure 1.** Location of the experimental station in relation to the spatial distribution of grasslands in Hulunbuir. Inset: Geographical location of the study area in China.

*2.2. Methods*

2.2.1. Experimental Setting

For this study, an experimental plot of 500 m by 320 m was established in Hulunbuir meadow steppe in 2006. A fence was installed on half of the plot, while the other half was left open and used in control experiments. The year 2006 was thus regarded as the first year of the enclosure. A quadrat random sampling design was adopted to determine sample point locations both inside and outside the fenced area (Figure 2). At each sample point, the height, coverage, density, and aboveground and belowground biomass of the plant community within the sampling quadrat were measured. A total of 20 quadrats were randomly chosen, half of which were inside the enclosed plot, and the other half was outside the enclosure. Each quadrat was 1 m × 1 m in size. Plants were sampled annually from 2006 to 2019 in early July when the plant growth was at the maximal level of the year. Samples collected in 2008, 2010, 2012, and 2014 (i.e., the third, fifth, seventh, and ninth year of enclosure; subsequently noted as F3, F5, F7, and F9, respectively) were used in this study.

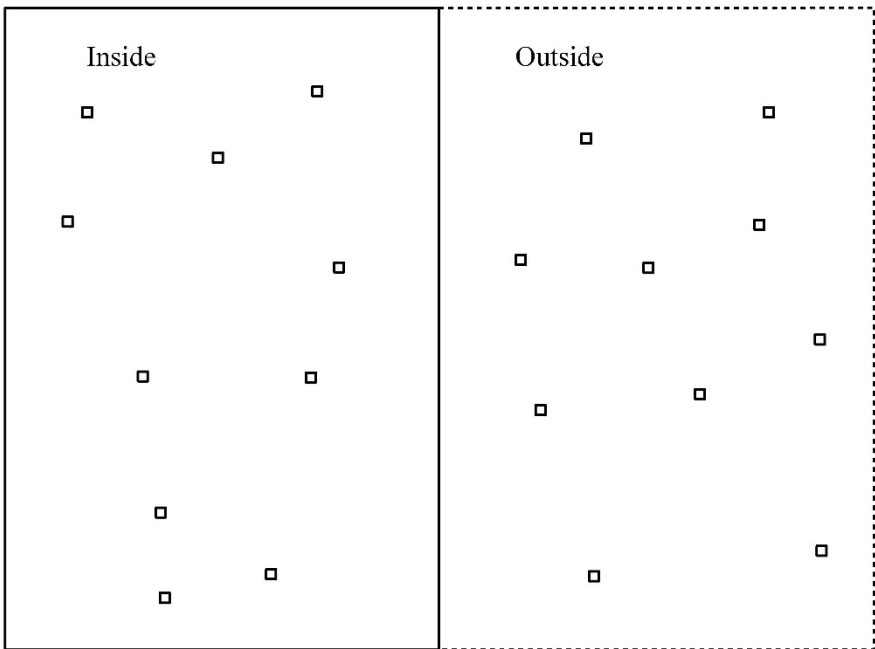

**Figure 2.** Layout of the quadrats in the experimental plot. A fence is installed along the border of the enclosure represented by the thick line, but not installed along the broken line which is outside the enclosure.

2.2.2. Measurements

The heights of ten randomly selected plants inside each quadrat were measured. The average height was derived from these measurements and used as the height of the plant community. Plant density was measured by counting the total number of plant individuals per quadrat. The area under vegetation cover and the area of bare soil inside each quadrat were visually estimated in-situ by experienced field staff, and the percent vegetative area was derived as the coverage of the vegetation community.

Aboveground biomass was evaluated by collecting and weighing the aboveground parts of the plants in each quadrat per species. Standing plants were cut at the soil surface per species and collected in sample bags. Fallen withered parts above the soil surface were also collected. Plant samples were separated into green parts and dead parts in the laboratory. The fallen withered parts were regarded

as dead parts. Collected green and dead parts were weighed to determine fresh weight per sample. Dry weight was also measured after the samples were oven-dried at 85 °C for 12 h.

Belowground biomass was determined by taking soil columns using a root drill (Beijing New Landmark Soil Equipment Co., Ltd, Beijing, China). Three soil columns were taken per quadrat at a depth range of 0–30 cm. The soil columns were water-washed using a 0.5 mm mesh net bag to retain plant roots in the laboratory. The obtained roots were oven-dried at 85 °C for 12 h before weighing.

*2.3. Diversity Indices*

The relative importance of a plant species in the plant community was evaluated using the plant importance value (PIV), which is given by the following equation [46]:

$$PIV = (RC + RH + RD + RB_a)/4,$$
(1)

where $RC$ is the relative coverage of a plant species; $RH$ is the relative height; $RD$ is relative density; and $RB_a$ is the relative aboveground biomass. Here, the values of $RC$, $RH$, $RD$, and $RB_a$ are defined as the percentage of a plant species' value of the coverage, height, density, and aboveground biomass to the total value of all plant species, respectively [46].

Plant diversity was evaluated using four indices, including the Shannon–Wiener diversity index [47], the Simpson diversity index [48], Pielou's evenness index [49] (pp. 89–93), and the Margalef richness index [50]. These indices are given by Equations (2) to (5), respectively.

$$H = -\sum (P_i \cdot \ln P_i),$$
(2)

$$D = 1 - \sum P_i^2,$$
(3)

$$J = H/\ln S,$$
(4)

$$d = (S-1)/\ln N,$$
(5)

where $P_i$ is the relative importance value of plant species $i$, $S$ is the total number of species sampled, and $N$ is the total number of plant individuals sampled.

*2.4. Data Analysis*

Data were processed and analyzed using Excel (Version 2013, Microsoft Corporation, Redmond, WA, USA) and SPSS (Version 19.0, IBM, NY, USA) packages. A one-way ANOVA was used to test for differences in community structure and diversity indices between the experimental years, while regression modeling was employed to analyze the relationships between above- and belowground biomasses, and between the plant's fallen withered parts and belowground biomass.

## 3. Results

*3.1. Plant Importance Values*

The importance values of sampled species both inside and outside the enclosure are summarized in Table 1. The importance values of the dominant species *L. chinensis* and the foundation species *S. baicalensis* were substantially higher inside the enclosure than outside the enclosure. In comparison, the importance values of the species *Artemisia frigida*, *Potentilla bifurca*, and *Carex duriuscula*, which are indicator species of grassland degradation, were lower inside the enclosure than outside the enclosure. The importance values of the other indicator species, *P. bifurca* and *C. duriuscula*, were found to follow similar patterns to species *A. frigida*.

The results of importance values of the species also showed that the importance of the dominant species *L. chinensis* inside the enclosure steadily increased with time and finally reached a maximum value of 30.87% in year F9. In contrast, as the enclosure duration increased, the importance values

of the degradation-indicator species *C. duriuscula* and *A. frigida* tended to decrease throughout the enclosure duration, reaching their minimum levels in year F9.

**Table 1.** The importance values of major plant species inside and outside the enclosure, %.

| Species | Enclosure Year | | | | | | | |
| --- | --- | --- | --- | --- | --- | --- | --- | --- |
| | F3 | | F5 | | F7 | | F9 | |
| | Inside | Outside | Inside | Outside | Inside | Outside | Inside | Outside |
| *Leymus chinensis* | 14.80 | 12.38 | 20.40 | 10.39 | 24.84 | 9.41 | 30.87 | 3.55 |
| *Stipa baicalensis* | 5.44 | 3.23 | 6.38 | 3.69 | 6.88 | 1.44 | 7.39 | 2.79 |
| *Heteropappus altaicuc* | 1.01 | 0.17 | 0.27 | 0.81 | 0.76 | - | 0.48 | - |
| *Thalictrum petaloideum* | 1.82 | 0.99 | 0.69 | 2.70 | 0.31 | 0.07 | 0.78 | 0.66 |
| *Melissilus ruthenica* | 5.64 | 2.23 | 0.49 | 0.96 | 0.03 | 1.25 | - | 1.34 |
| *Bupleurum scorzonerifolium* | 1.86 | 0.68 | 0.73 | 1.44 | 0.96 | - | 0.77 | 0.47 |
| *Artemisia frigida* | 1.14 | 1.61 | 1.07 | 1.18 | 0.25 | 0.45 | 0.11 | 1.66 |
| *Potentilla bifurca* | 0.92 | 1.00 | 0.27 | 2.08 | 1.26 | 2.47 | 1.30 | 1.93 |
| *Artemisia laciniate* | 7.99 | 9.34 | 7.86 | 7.12 | 8.98 | 0.68 | 7.34 | 5.26 |
| *Serratula komarovii* | 7.21 | 3.54 | 4.98 | 5.02 | 3.73 | 0.91 | 6.18 | 2.17 |
| *Galium verum* | 1.90 | 1.25 | 4.22 | 1.51 | 2.40 | - | 2.10 | - |
| *Carex pediformis* | 11.56 | 10.70 | 6.28 | 9.72 | 11.12 | 0.27 | 12.71 | 3.00 |
| *Adenophora stenanthina* | 2.54 | 1.95 | 4.32 | 2.13 | 3.08 | 0.31 | 3.90 | 1.28 |
| *Carex duriuscula* | 2.01 | 9.67 | 9.60 | 10.78 | 4.59 | 46.06 | 0.63 | 20.89 |
| *Palsatilla turczaninovii* | 6.47 | 3.99 | 6.69 | 11.35 | 2.82 | 0.16 | 2.93 | 4.69 |
| *Cleistogenes squarrosa* | 1.64 | 5.27 | 3.44 | 2.68 | 0.45 | 4.19 | 0.48 | 11.31 |
| *Achnatherum sibiricum* | 2.15 | - | 1.29 | 1.02 | 1.84 | 2.44 | 5.13 | 4.39 |
| *Iris ventricosa* | 2.13 | 1.73 | 0.29 | 1.08 | 2.41 | 0.30 | - | - |
| *Thalictrum squarrosum* | 1.81 | 1.02 | 5.27 | 3.30 | 8.01 | 0.06 | 3.66 | - |
| *Koeleria cristata* | - | 1.84 | - | 3.44 | 1.26 | 0.19 | - | 3.54 |

### 3.2. Community Height, Coverage, and Density

The variations of plant height, vegetation coverage, and plant density in relation to the enclosure duration are given as bar charts in Figure 3. A comparison between the height of the plants inside and outside the enclosure showed that plants inside the enclosure were significantly higher than plants outside the enclosure. In the third year of the enclosure (F3), for example, plants inside were 189.50% higher than plants outside. After 9 years of the enclosure (F9), plants inside reached a height which was 4.5 times that of the plants outside the enclosure (Figure 3a). Overall, plant height inside the enclosure showed a slight decreasing trend until the fifth year and then a more prominent increasing trend, although statistical significance was not attached. The plant height inside the enclosure maximized in year F9, which was 32.26%, 47.17%, and 13.11% higher than previous years, respectively.

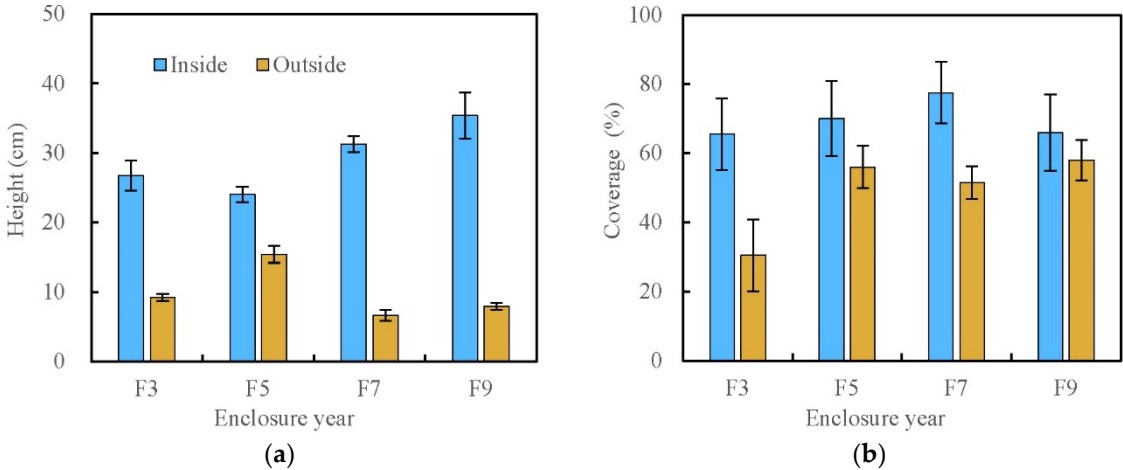

(a)　　　　　　　　　　　　　　　　　　　　　　　　(b)

**Figure 3.** *Cont.*

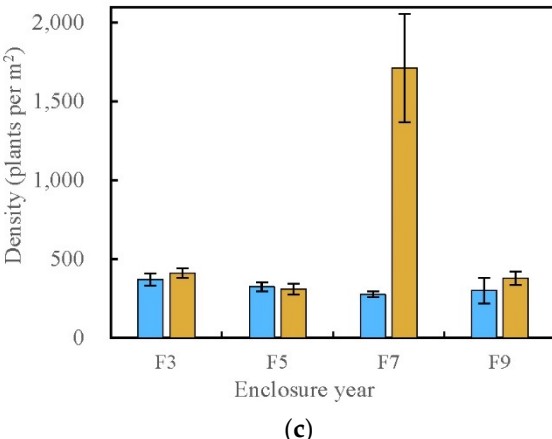

**(c)**

**Figure 3.** Variations in plant community characteristics observed both inside and outside the fenced enclosure in the third, fifth, seventh, and ninth years of the enclosure. Observed community characteristics include (**a**) species height in centimeters, (**b**) species' relative ground coverage in percentage, and (**c**) plant density in terms of count of plant individuals per square meter.

The enclosure also had a significant effect on the vegetation cover of the plant community. The plant coverage inside the enclosure was observed to be higher than that outside the enclosure by a margin of 115%, 25%, 50%, and 14% in years F3, F5, F7, and F9, respectively. As the duration of the enclosure increased, the coverage showed a temporal trend of first increasing and then decreasing, and the maximum coverage value was observed in year F7 (Figure 3b).

No significant effect of the enclosure was found on plant density (Figure 3c), although density inside the enclosure was significantly lower than that outside the enclosure in year F7.

*3.3. Community Diversity*

The variations of the Shannon–Wiener diversity index, the Simpson diversity index, the Pielou's evenness index, and the Margalef richness index values in relation to the enclosure years are shown in Figure 4. The Shannon–Wiener, Simpson, and Margalef indices inside the enclosure were all observed higher than their values outside the enclosure in years F3, F5, and F7. The inside–outside difference of these indices was tested statistically significant in year F7, while significance was not found for the difference in other years. Moreover, only slight inside–outside difference was observed in the Pielou's evenness index values in years F3 and F5 (Figure 4d). In year F9, all the measurements, regardless of diversity, evenness, or richness, showed lower values inside the enclosure than outside the enclosure, albeit not significant.

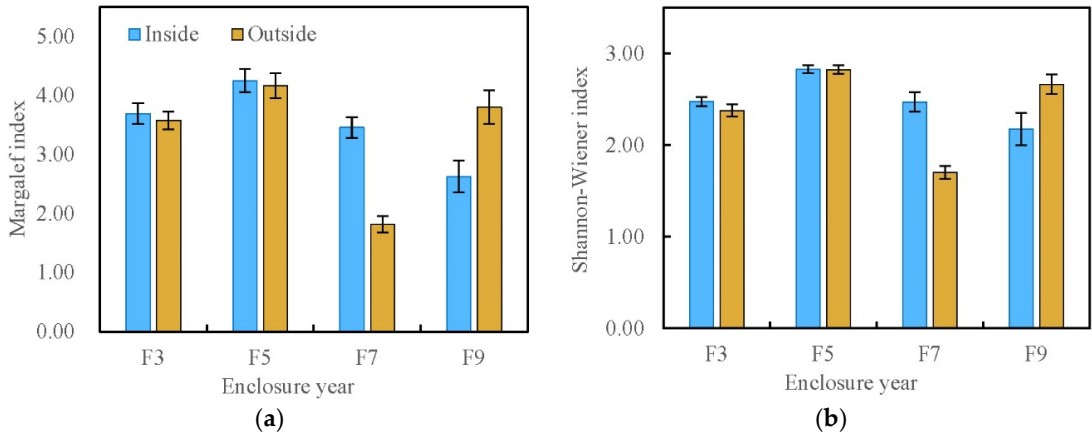

**(a)** **(b)**

**Figure 4.** *Cont.*

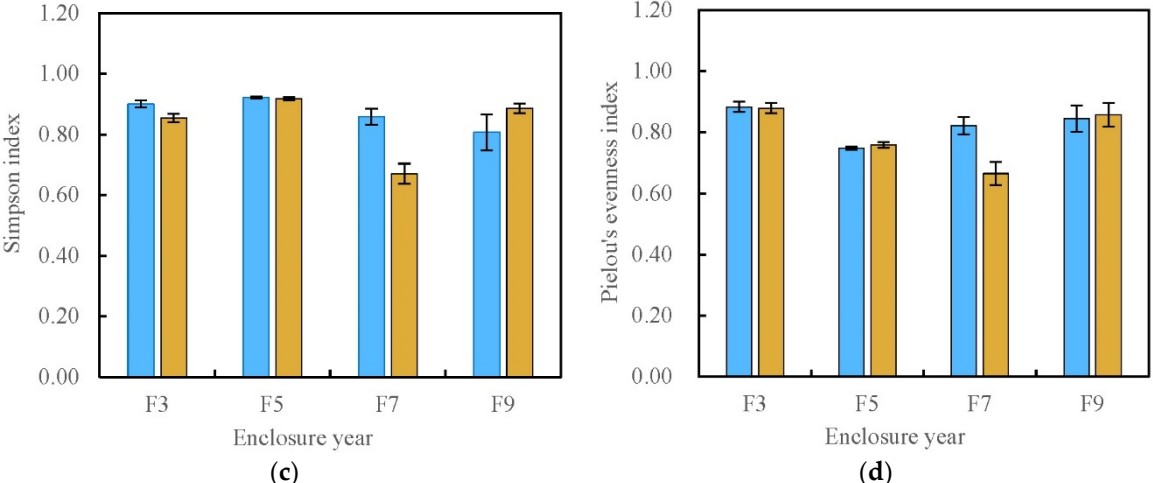

**Figure 4.** Temporal variations in community diversity of the temperate meadow steppe in Hulunbuir as observed in the plot experiment. Grass species both inside and outside the enclosure were under observation. Multiples indices of plant diversity were derived, including (**a**) Margalef richness index, (**b**) Shannon–Wiener diversity index, (**c**) Simpson diversity index, and (**d**) Pielou's evenness index.

It was notably observed that the Shannon–Wiener and the Simpson diversity indices, together with the Margalef richness index, showed a synchronous pattern whereby these indices tended to increase first and then decrease. All three indices reached a maximum level in year F5 which acted as a turning point. It is interesting to indicate that the Pielou's evenness index showed an opposite trend. The Pielou's index decreased first and then increased, with a turning point also in year F5.

### 3.4. Biomass Production

The relationship between plant biomass and the enclosure duration is given in Figure 5. The plants inside the enclosure produced higher aboveground biomass than plants outside the enclosure throughout the experimental period (Figure 5a). An inside–outside advantage was achieved at 224.16%, 66.00%, 176.31%, and 309.19% in years F3 through F9, respectively. The general trend was that the aboveground biomass increased with time, reaching the maximum production in year F9. The maximal aboveground biomass in year F9 was higher than in previous years by 65.22%, 46.31%, and 2.56%, respectively, showing a decreasing growth rate.

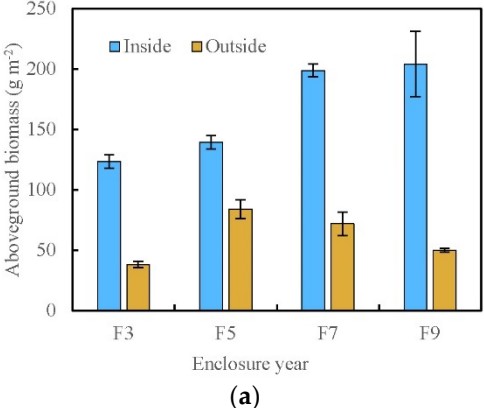

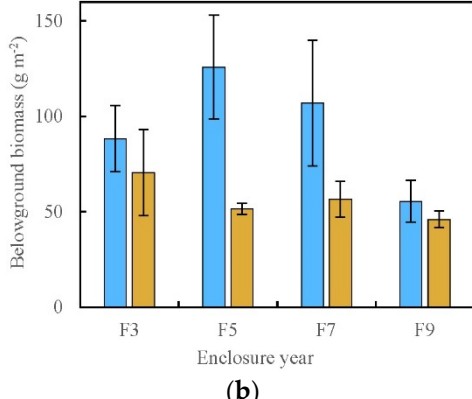

**Figure 5.** *Cont.*

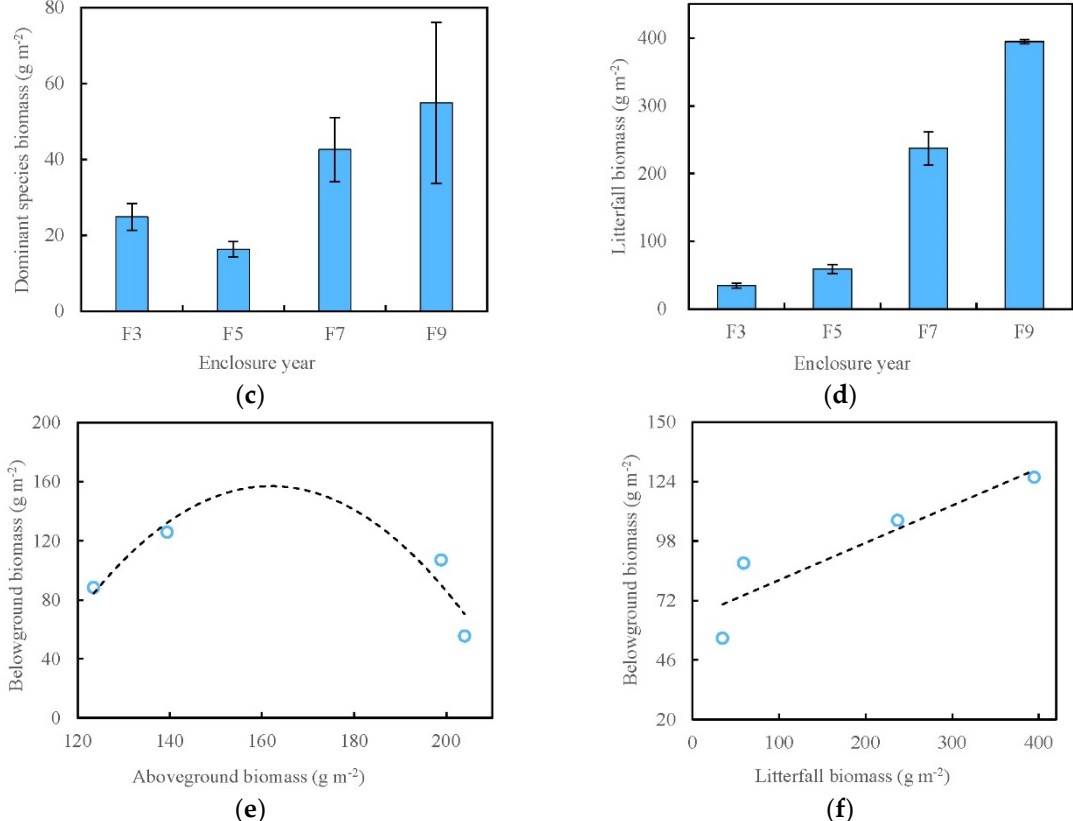

**Figure 5.** Temporal variations in plant biomass and relationships between biomasses measured above and below the soil surface. Aboveground biomass (**a**) and belowground biomass (**b**) were measured both inside and outside the enclosure, but the aboveground biomass of the dominant species (**c**) and the litterfall biomass (**d**) were only measured inside the enclosure. (**e**) Quadratic relationship between aboveground biomass and belowground biomass: $y = -0.05x^2 + 15.91x - 1131.50$ ($R^2 = 0.79$, $p = 0.001$) and (**f**) linear relationship between litterfall biomass and belowground biomass: $y = 0.12x + 116.65$ ($R^2 = 0.90$, $p = 0.001$).

A similar advantage also existed in belowground biomass under the enclosure treatment (Figure 5b). The belowground biomass measured from soils inside the enclosure was found consistently higher than that outside the enclosure. The inside–outside margin was measured at 25.24%, 143.98%, 89.02%, and 20.47% in years F3 though F9, respectively. Different from the all-time growth pattern in aboveground biomass, the belowground biomass showed a trend of first increasing and then decreasing with a turning point in year F5. Belowground biomass in year F5 was higher than years F3, F7, and F9 by 42.44%, 17.56%, and 126.67%, respectively. Similar increasing trends to that of the aboveground biomass were observed in the biomass of dominant plant species and in the litterfall biomass, both inside the enclosure (Figure 5c,d).

Cause–effect relationships were statistically established from regression analyses involving the belowground biomass versus the aboveground and the litterfall biomasses (Figure 5e,f), showing a quadratic and a linear relationship between the above- and belowground biomass yields and between litterfall and belowground biomass yields, respectively.

## 4. Discussion

### 4.1. Plant Community Characteristics

Height, coverage, and density are important characteristics of grassland plant communities [11,13,51,52]. Vegetation coverage and plant density reflect the strength of photosynthetic

capacity of green plants, lushness of vegetation cover, and potential of economic traits [9,10]. Our results showed that enclosure caused substantial increases in plant coverage and height. By growing under a condition without cattle grazing and human disturbances, the plants were able to complete a normal life cycle of growth, flowering, setting seeds, etc. Higher quantities of branches, roots and taller plants were consistently observed in the experimental plot. Compared to plants outside the enclosure, the height and coverage of the plant community inside the enclosure were greatly increased (Figure 3a,b). Some tall grasses such as *L. chinensis* are palatable food for cattle, the dominant livestock species in the region. Consequently, the competitive advantage of these tall grasses was effectively suppressed by livestock grazing in open steppe. As such, the shorter species such as *C. duriuscula* obtained more space to grow outside the enclosure, which gave rise to higher density and higher dominance of these species, as confirmed by our results (Figure 3c). Overall, the results obtained from our experimental plot are in line with other works conducted in the region [6,8,12,13,22] or elsewhere under similar conditions [7,17,30,53].

### 4.2. Plant Diversity

Plant diversity is one of the most critical factors in maintaining the development and productivity of grassland ecosystems [54]. Previous studies on the response of the plant diversity to the enclosure duration have produced mixed results. While some authors (e.g., [55]) suggested that the fence enclosure increased species diversity, others indicated that enclosure had mixed effects on plant diversity [41]. As Cardinale et al. [56] argued, in addition to the enclosure duration, differences in grassland type, habitat, and grazing intensity may all contribute to the differential observational outcomes across the border of the enclosure. In other words, observed differences in community diversity were not necessarily caused by the enclosure treatment alone. In the beginning of our study, species richness and diversity of the enclosed temperate meadow steppe, which had endured severe degradation for years, increased as compared with the open grassland. Even under fence protection, the plant community inside the enclosure did not show a constant upward trend in species richness and diversity. Conversely, it showed a brief increase in species richness and diversity under the enclosure until year F5, followed by a longer period of decrease (Figure 4). There is no doubt that the enclosure prevents livestock from entering and thus creates a beneficial environment with less soil compaction as a direct result of temporary absence of grazing [27,40] and other disturbances [55]. Moreover, enclosure conserves the soil seed bank [36,41], which in turn contributes to the species richness and diversity. However, as the enclosure protection lasts longer in time, the dominant species in the plant community becomes more competitively superior, creating an increasingly tougher environment for the weaker or rarer species due to the so-called compensatory effect [57]. As a result, the species richness and diversity were displaying a downward trend from year F5 to F9 (Figure 4a–c). In contrast, as the enclosure duration increased, the evenness of the plant species in the community first decreased and then increased. It was found that high levels of richness and diversity in the community species led to a high degree of habitat partitioning [34], suggesting that species evenness tended to increase with the enclosure duration (Figure 4d).

### 4.3. Grassland Productivity

Biomass refers to the total amount of material production, which includes plant parts above the surface and plant roots below the surface. Plant roots play a pivotal role in mediating ecosystem functioning [58]. Moreover, roots bridge the energy and nutrient exchange between above- and belowground ecosystems [59,60]. Biomass is an effective measurement of plant productivity and performance; it is also one of the most basic quantitative characteristics of the grassland ecosystem [61,62]. While aboveground biomass indicates the plant community's ability to accumulate organic matter over a specific duration of time [24], belowground biomass depicts the grassland ecosystem's functioning to input carbon and other critical elements [63]. Our results (Figure 5a–c)

showed significantly positive effects of the enclosure on either aboveground, belowground, or dominant species biomass, which was largely in agreement with the results obtained elsewhere [52,53].

Although studies have demonstrated that enclosures could substantially increase aboveground biomass, the duration of grassland enclosures cannot be infinite, given the principles of sustainability for grassland ecosystems [57]. Excessively long duration of enclosure may produce unwanted, adverse effects on the normal growth and development of forage grasses. Unharvested hay piling on soil surface will likely inhibit plant regeneration and seedling growth, thus hindering grassland renovation. It has been reported that proper mowing or grazing tends to encourage the tillering and regeneration of forage grasses and, therefore, to improve the overall quality of grassland [64].

### 4.4. Optimal Enclosure Duration

Ecological restoration has been defined by the Society of Ecological Restoration International (SER) as the process of assisting the recovery of an ecosystem that has been degraded, damaged, or destroyed [65]. Similarly, the restoration of a degraded grassland can be defined as the process of helping the grassland recover from the degradation status. The recovery can be measured using resilience, stability, and performance attributes of the grassland. Accordingly, the grassland attributes of plant diversity, vegetation coverage, and biomass can be used as proxies of grassland resilience, stability, and performance, respectively [65]. As discussed above, plant diversity, no matter if measured by the Margalef, the Simpson, or the Shannon–Wiener index, reached a maximum value in the fifth year of the enclosure (Figure 4), while the maximum value of the vegetation coverage was reached in the seventh year of the enclosure (Figure 3b). In addition, as an indication of the economic yield of the grassland, aboveground biomass kept an increasing trend at least during the first nine years of the enclosure; however, its growth rate reached a maximum in the seventh year and began to drop from the ninth year onwards (Figure 5a). Moreover, regarded as the energy and nutrient exchange media between the above- and belowground terrestrial ecosystems [58–60], the temporal variation pattern of belowground biomass deserves extra attention. Belowground biomass increased since the beginning of the enclosure and reached a maximum in the fifth year and then began to decrease (Figure 5b). The decreasing trend is so prominent that until the ninth year of the enclosure, the accumulated belowground biomass was less than 50% of the maximum value in the fifth year. Overall, judged on the basis of the variation patterns of these grassland attributes, the ecosystem conditions of the temperate meadow steppe in the study area responded positively to the enclosure treatment. The conditions of this degraded temperate meadow steppe were steadily improved as the enclosure duration lasted. An optimal condition was achieved after approximately 5–7 years under the protection of the enclosure.

## 5. Conclusions

We conducted a multi-year field experiment to investigate the effects of a fence enclosure on the improvement in quality and wellbeing of a heavily degraded temperate meadow steppe in Hulunbuir in northern China during 2006–2019. Our obtained results clearly show that the fence enclosure is an efficient restoration measure that not only promotes species diversity but also improves grassland productivity. Our obtained results also show that although a degraded grassland can recover steadily under the protection of a fenced enclosure, it takes at least five to seven consecutive years before the grassland vegetation reaches a somewhat stable condition in terms of plant health and productivity. These seemingly simple results have profound technical and policy implications for grassland management and livestock development in the region and beyond.

First of all, our results send a clear warning message to the grassland communities and to the decision-makers that any optimism over the effects of degradation control programs executed recently across the region is likely premature and thus needs to be cautioned. Driven by strong market demand, the grasslands of Inner Mongolia have seen dramatic eco-environmental changes in recent decades. While the livelihood of the grassland communities has largely been improved, grassland degradation and desertification have become more and more prominent. Due to increasing numbers of livestock and

changed land use from semi-nomadic systems to a sedentary system, size and productivity of typical steppes in the study region have substantially decreased [66]. In response, the government initiated and implemented a series of grassland restoration programs, such as the Natural Grassland Restoration Program and the Returning Grazing Land to Grassland Program [5]. Meanwhile, a range of policy reforms have been instituted since the beginning of the 21st century, which included the Grassland Eco-compensation Program, the Farmer's Professional Cooperative Law, and so on. Preliminary satellite monitoring data suggested that these programs and institutions produced a significant and positive effect in mitigating degradation rate and accelerating grassland rehabilitation, especially in the pilot counties [18]. However, do these short-term gains necessarily lead to a long-term success? Our data contribute to the scientific understanding of this discussion by proving biophysical evidences on the grassland restoration process. Our experiments show that a degraded grassland has been very responsive to protection measures since the very beginning when grazing disturbances were removed. However, the protection measures need to be maintained and kept in place continuously for a duration of approximately 5–7 years before the grassland ecosystem begins to stabilize. Any improvements from shorter-period intervention measures will be short-lived, too. There is no doubt that the long-term goals of grassland conservation cannot be realized by short-term actions.

Secondly, our experimental evidence also shows that the absence of grazing disturbances is by itself an effective grassland restoration measure. As such, any technical interventions that promote pastural fallowing and mitigate overgrazing should be encouraged. These include, among others, desertification control [21,67], grazing prohibition [5], grazing rotation [68], and artificial grassland plantation [7,30]. Moreover, in order to meet increasing needs of livestock products, the pattern of rearing livestock in captivity, which relies on imported forage as a feed source and lessens the pressure on local grassland, should be further encouraged. In line with the feed demand, expansion of cultivated pasture establishments should be further facilitated [5,18].

Thirdly, policy reforms and institutional changes are needed to provide guidance and support for technical interventions. On the one hand, institutional reforms to reinforce the long-term grassland use rights are needed. Recent policy adjustments to the household responsibility system known as the 'separating three property rights' (STPR) scheme had detrimental effects on grassland protection and conservation. Although STPR was designed to promote grassland transfers [17] and thus to facilitate the economic development in pasture areas, grassland tenants who obtained land use rights under STPR tended to seek short-term gains rather than longer-term sustainable productivity. To cope with this issue, solutions such as financial incentives to stimulate grassland protection [69] and payment for ecosystem services (PES) [70] are preferred. For example, PES can be used as an active intervention to compensate a herder's economic losses caused by putting degraded grasslands into fallowing or by lowering the stocking rate once a carrying capacity threshold is passed [4,71]. On the other hand, trade liberalization of livestock products could help reduce grassland use intensity. Exploring the potential to increase the productivity of the domestic livestock is another way to address this concern, for instance, through the diffusion of techniques for feeding and fattening and by promoting the use of improved breeds [72].

Lastly, we would like to point out that innovations are needed in the current thinking of grassland conservation in China. For example, introduction of greater bottom-up decision-making may be considered at the local level to cope with time-sensitive operations (e.g., rotational grazing) more effectively according to changing weather conditions [73]. Bottom-up decision-making is also needed in situations of, e.g., the prevention of "night-grazing", a rampant illegal action in recent years that breaches grazing prohibition at nighttime when patrol is off duty [74]. Furthermore, involving community-level officials in the co-management of the trading of the grassland use rights, as demonstrated by a Tibetan pasture community in its customary management of grassland and livestock [17], is yet another example of innovative thinking. As a matter of fact, the technical options recommend here are more relevant at the local scale. Without bottom-up decision-making, the feasibility of these options is hardly

imaginable. Above all, a balanced livestock–grassland system based on local conditions needs to be established to pave the way for sustainable grassland utilization and livestock production [7].

**Author Contributions:** Conceptualization, L.X. and X.X.; investigation, L.X. and Y.N.; methodology, L.X. and G.Y.; data curation, B.C., Y.N. and G.Y.; formal analysis, Y.N. and G.Y.; writing—original draft preparation, L.X. and L.Y.; writing—review and editing, L.Y.; visualization, L.Y. and D.X.; supervision, L.Y.; project administration, X.X. and G.Y.; resources, G.Y. and B.C.; funding acquisition, X.X. and L.X. All authors have read and agreed to the published version of the manuscript.

**Funding:** This research was funded by National Natural Science Foundation of China (No. 41703081), National Key Research and Development Program of China (2016YFC0500603, 2017YFC0503805, 2017YFE0104500), China Agriculture Research System (CARS-34) and National Nonprofit Institute Research Grant of CAAS (912-32).

**Acknowledgments:** Logistic supports from Ghent University for this paper are kindly acknowledged.

**Conflicts of Interest:** The authors declare no conflict of interest.

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
