# Peer review of "Effects of Fence Enclosure on Vegetation Community Characteristics and Productivity of a Degraded Temperate Meadow Steppe in Northern China"

_applsci, doi:10.3390/app10082952_

Round 1

Reviewer 1 Report

I appreciate the opportunity to review this interesting report on the "Effects of fence enclosure on vegetation community characteristics and productivity of a degraded temperate meadow steppe in northern Chinas". I enjoyed reading the manuscript. I commend the authors for a number of strengths of their work, including:

 1. The research question is one of the most important parts of your research. What is the main question addressed by the research?
The paper fails to make a clear hypothesis and works on a hypothesis that has already been explored and established.

2. The review of the literature is not thorough so the reader is not given an adequate background about the topic.

3. Are the conclusions consistent with the evidence and arguments presented? Do they address the main question posed? The arguments and data presented are not complete in establishing the core idea presented

4. The results observed can be used as indicators of exclosure effectiveness and provide a foundation for sustainable management.
Authors can improve the introduction part and discuss it.

5. The English in the present manuscript is not of publication quality and require major improvement. Please carefully proof-read spell check to eliminate grammatical errors.

Reviewer 2 Report

see attachment

Round 2

Reviewer 1 Report

The authors have shown efforts to improve the manuscript and this should be well appreciated.

English language and style are minor spell check required. Please carefully proof-read spell check to eliminate grammatical errors.

I.e. :

L 20,23, 24, ...: change 'enclosur' to 'the  enclosur'

L 24: change 'experimental' to 'the experimental'

L 37: '38 ,'  change to '38,'

and ....

Author Response

Dear Reviewer,

Thank you for your careful review and your positive evaluation of our revised manuscript. We accepted your language corrections with appreciation and implemented them across the text in places of your indication and in similar occurrences. 

We detail our responses here:

  • L 20,23, 24, ...: change 'enclosur' to 'the enclosur'

Yes, we changed "enclosure" to "the enclosure" in these lines and other lines across the maintext: L 117, 119, 128, 158, 300, 301, 321, 324, 327, 329, 628, 629, 630, 632, 638, 639, 709.

  • L 24: change 'experimental' to 'the experimental'

Yes, we change it as instructed in L24 and in a similar appearance in L211/212.

  • L 37: '38 ,' change to '38,'

We did not find this text in L37. Nevertheless, we went through the entire manuscript to ensure that it did not contain similar errors.

  • and ....  

Again, we double-checked the manuscript and corrected all similar errors: 

L18 "fence enclosure" changed to "the fence enclosure";
L82 "the duration of enclosure engagement" changed to "the duration of the enclosure engagement";
L96 "enclosure treatment" changed to "the enclosure treatment";
L100 "enclosure" changed to "an enclosure";
L113 "fence enclosure" changed to "a fenced enclosure";
L153 "Fence" changed to "A fence"; "enclosure" changed to "the enclosure".

Thank you again for your swift and accurate review!

Sincerely yours,

The authors, 20 April 2020